# Preparation of Bioactive De-Chlorophyll Rhein-Rich *Senna alata* Extract

**DOI:** 10.3390/antibiotics12010181

**Published:** 2023-01-16

**Authors:** Wah Wah Aung, Kanokpon Panich, Suchawalee Watthanophas, Sutada Naridsirikul, Juthaporn Ponphaiboon, Wantanwa Krongrawa, Pattranit Kulpicheswanich, Sontaya Limmatvapirat, Chutima Limmatvapirat

**Affiliations:** 1Department of Industrial Pharmacy, Faculty of Pharmacy, Silpakorn University, Nakhon Pathom 73000, Thailand; 2Pharmaceutical Biopolymer Group (PBiG), Faculty of Pharmacy, Silpakorn University, Nakhon Pathom 73000, Thailand; 3Baan Teraeng Groups Co., Ltd., 12/32 Moo 19 Chumhed, Muang Buriram, Buriram 31000, Thailand

**Keywords:** *Senna alata*, rhein, ultrasound-assisted extraction, Box–Behnken design, biological activity

## Abstract

*Senna alata* leaves display various biological activities as a result of their rhein and phenolic composition. The objective of this study was to develop bioactive de-chlorophyll rhein-rich *S. alata* extracts. The rhein content was quantified using a validated high-performance liquid chromatography–diode array detection (HPLC–DAD) method. The best process parameters for maximizing rhein were established using ultrasound-assisted extraction (UAE). The optimal conditions for the parameters were determined using the Box–Behnken design (BBD); 95% *v*/*v* ethanol was used as the extraction solvent at 59.52 °C for 18.4 min with a solvent-to-solid ratio of 25.48:1 (mL/g) to obtain the predicted value of rhein at 10.44 mg/g extract. However, the color of the rhein-rich extract remained dark brown. For the removal of chlorophyll, liquid–liquid extraction with vegetable oils and adsorption with bleaching agents were employed. The bleaching agents were significantly more effective at removing chlorophyll and had less of an effect on the reduction in rhein content than vegetable oils. The presence of rhein and phenolics in the de-chlorophyll extracts might be responsible for their antioxidant, anti-inflammatory, and antibacterial activities. These findings indicate that rhein-rich extract and its de-chlorophyll extracts possess sufficient biological activities for the further development of cosmeceuticals and pharmaceuticals.

## 1. Introduction

It has been discovered that rhein (4,5-dihydroxyanthraquinone-2-carboxylic acid) (Figure 1), a lipophilic anthraquinone contained in *Senna alata* (also known as *Cassia alata*), protects hepatic injury and fibrosis, inhibits renal interstitial fibrosis, lowers cholesterol and liver triglyceride levels, increases the synthesis of type II collagen and aggrecan in osteoarthritic chondrocytes, reduces inflammation-induced vascular complications, suppresses the production of reactive oxygen species, restrains the cell cycle and viability of breast cancer cells, reduces fasting blood glucose levels, improves glucose tolerance and inhibits various *Staphylococcus aureus* strains [1,2,3,4,5]. All the aforementioned pharmacological effects in a variety of experimental models have substantiated that rhein possesses antihepatotoxic, nephroprotective, lipid-lowering, chondroprotective, anti-inflammatory, antioxidant, antiproliferative, anti-diabetic and antibacterial activities, providing support for several of its potential therapeutic properties. Due to pharmacological activities, rhein has emerged as a promising chemical for treating a number of skin problems [6]. The antioxidant, anti-inflammatory and antimicrobial properties of the rhein-rich *S. alata* extract, which can be developed as an active component in cosmeceuticals and pharmaceuticals, are particularly remarkable.

Cosmetics with active ingredients with drug-like qualities are referred to as “cosmeceuticals”. Cosmetics with therapeutic qualities have positive local effects and prevent skin disorders from progressing [7,8]. They provide the bioactive compounds necessary for healthy skin, which improve beauty, can lessen wrinkles and enhance the brightness, texture and tone of the skin. Herbal extracts have long been employed in cosmeceuticals and pharmaceuticals; natural bioactive compounds, in particular, are safer than synthetic compounds [9]. According to comprehensive studies, plant rhein and polyphenols contain antioxidant and anti-inflammatory capabilities. Numerous studies on human skin and skin cell lines have demonstrated the advantages of rhein and polyphenols; as a result, these substances are being added to an increasing number of cosmeceuticals and pharmaceuticals [10,11,12].

Extraction conditions are crucial for obtaining the desired yield and quality of plant extracts, including phytochemical constituents and biological properties. To maximize the content of active components, it is critical to select the best extraction method for each plant. For *S. alata* leaf extracts to be more effective, they should have a high concentration of rhein. Conventional extraction has time, energy and solvent limitations. However, ultrasound-assisted extraction (UAE) can successfully extract many bioactive compounds in less time with less energy and solvent. Non-thermal extraction methods such as UAE retain more of the bioactive chemicals’ effectiveness. It is crucial to understand and manage the variables associated with UAE, such as frequency, power, temperature, time, solvent type, solvent concentration and solvent: solid ratio [13,14,15]. The efficiency of bioactive compound extraction is determined by the degree of disruption of the plant leaves. Scanning electron microscopy analysis revealed substantial contraction of epidermal and mesophyll cells of plant leaf tissue in UAE compared to that of maceration extraction, which yielded a higher content of bioactive compounds than that of maceration extraction [16]. The majority of earlier investigations have used methanol as an extraction solvent to prepare *S. alata* leaf extracts [17,18,19]. The poisonous nature of methanol is well recognized. Nontoxic extraction solvents, such as ethanol, water or their binary combinations, which are known to be safe for human ingestion, are believed to be effective solvents for the extraction of bioactive herbs [5,20,21]. The manufacturing of cosmeceuticals and pharmaceuticals containing *S. alata* leaf extract has recently faced significant difficulties because of the dark color and low rhein concentration of the extract. Therefore, in this investigation, ethanol and distilled water were used as extraction solvents in UAE to improve the concentration of rhein in the *S. alata* leaf extract. Furthermore, to reduce the dark color, interfering compounds in the extract, particularly chlorophyll, should be removed.

The use of an experimental design for evaluating and optimizing extraction processes enables the collection of the most useful information from fewer experiments, thereby minimizing costs and maximizing desired outcomes. Response surface methodology (RSM) is an effective statistical technique for the development of an empirical model, employing the most influential variables and effects [22,23]. The Box–Behnken design (BBD), an RSM technique, has been demonstrated to be effective for optimizing herbal extraction [24]. The optimization design yields optimal conditions with a modest number of experimental trials. BBD only requires three levels of each factor and a limited number of experimental runs. In addition, the literature that investigated how to increase rhein content via UAE in the leaves of *S. alata* has not mentioned this.

One of the difficulties of developing a plant-based formulation is the intense color of the extracts. This may result in products having an unappealing appearance. The major causes of the intense dark color of the herbal extracts are chlorophylls, which make them unsuitable as an ingredient in cosmeceutical and pharmaceutical products despite their beneficial properties [25]. As a result, there is an urgent need to develop a method for removing the intense color of herbal extracts while retaining their high biological effects. Alternatively, colored compounds from herbal extracts could be removed using an adsorption process. A variety of decolorizing agents, such as aluminum oxide and activated charcoal, are currently used to remove colored compounds [26]. Decolorization is a key step for giving a suitable physical appearance due to the high level of chlorophyll-type chemicals found in *S. alata* leaf extracts. Nonetheless, the extract’s dark brown color causes challenges in the development of cosmeceuticals and pharmaceuticals. In this investigation, chlorophyll in *S. alata* leaf extract was removed using a safe and effective approach. The extract from which chlorophyll was taken out for this study is called “de-chlorophyll extract.” 

Generally, aqueous ethanol has been used to prepare edible, polyphenol-rich extracts from plant materials [27]. However, the co-extraction of high chlorophyll concentrations occurs when organic solvents are used [28]. Chlorophyll-containing plant extracts have the potential to alter the final product’s hue [29]. It is common knowledge that chlorophyll can be reduced or eliminated by adsorption with activated charcoal or liquid–liquid extraction with various organic solvents (acetone, propanol and chloroform), as employed in the commercial plant extraction industry [25,30]. However, these methods are not completely selective, and decolorized extracts can lose bioactive compounds as a result. The residue of those organic solvents that might harm the human body is another key issue with the preparation of plant extracts for use in the manufacture of cosmeceuticals and pharmaceuticals [31]. In accordance with the “like dissolves like” concept, liquid–liquid extraction with vegetable oils can be safe and free of solvent residues in the resulting extract. Furthermore, palm and coconut oils are human-friendly and contain nutrients for healthy skin [32]. In this investigation, liquid–liquid extraction with vegetable oils (palm oil or coconut oil) and adsorption with bleaching agents (aluminum oxide or activated charcoal) were chosen as a secure way to remove chlorophyll from the rhein-rich *S. alata* extract.

Among the analytical methods for determining phytochemicals, high-performance liquid chromatography–diode array detection (HPLC–DAD) is one of the most often used methods due to its lower cost and excellent accuracy and precision for routine analysis [33]. In previous research, rhein concentrations in extracts of *Cassia fistula* fruit pulps were determined using high-performance liquid chromatography–ultraviolet/visible detection (HPLC–UV/Vis) with a reverse phase (RP)-C18 column, a mobile phase of methanol of 0.5% *v*/*v* aqueous acetic acid (60:40) and a detection wavelength of 435 nm [34]. Analysis of rhein in *S. alata* leaf extracts via HPLC–DAD requires an RP-C18 column with a mobile phase of methanol of 2% *v*/*v* aqueous acetic acid (70:30) and detection at 254 nm [35]. Due to its examination of peak purity, DAD is superior to UV/Vis. Peak purity analysis is a method for detecting coeluting contaminants on HPLC chromatograms in order to ensure accuracy and avoid the formation of misleading analytical data. It is thus a helpful supplement to standard quality control procedures, especially in pharmaceutical and food analyses [36].

This study aimed to produce a rhein-rich *S. alata* leaf extract and its de-chlorophyll extracts with biological activities, as well as to develop an HPLC–DAD method for rhein determination. The research involved five steps: (1) optimizing the conditions for rhein extraction from *S. alata* leaves and producing multiple *S. alata* leaf extracts using the UAE method; (2) developing and validating the HPLC–DAD method for assessing rhein contents in obtained extracts; (3) examining the yields, rhein contents, total phenolic contents and antibacterial activities of the obtained extracts; (4) removing chlorophyll from the rhein-rich extract using a green method, such as bleaching agents and liquid–liquid extraction with vegetable oils; and (5) investigating the rhein contents, total phenolic contents, antioxidant activities, anti-inflammatory activities and antibacterial activities of de-chlorophyll extracts prepared from the rhein-rich extract.

## 2. Materials and Methods

### 2.1. Materials and Reagents

The Thai Herbal Pharmacopoeia [37] was used to identify the *S. alata* leaves that were cultivated in an organic garden in Chumhed, Muang Buriram, Buriram, Thailand. A preserved specimen of the plant was then placed in the Faculty of Pharmacy, Silpakorn University. Both the rhein analytical standard (purity ≥ 95% *w*/*w*) and the 2,4,6-tripyridyl-s-triazine (TPTZ) were acquired from MilliporeSigma Supelco (Darmstadt, Germany). VWR International (Fontenay-sous-Bois, France) and Ajax Finechem (Auckland, New Zealand) provided ethanol absolute (≥99.8% *v*/*v*) and orthophosphoric acid (85% *w*/*w*), respectively. Methanol HPLC grade (99.8% *v*/*v*) and acetonitrile HPLC grade (99.9% *v*/*v*) were purchased from Fisher Scientific Korea Ltd. (Seoul, Korea). Greater Pharma Manufacturing Co., Ltd. (Nakhon Pathom, Thailand) supported the neomycin sulfate standard. Diclofenac sodium (Department of Medical Sciences Reference Standards, DMScRS) was bought from the Bureau of Drug and Narcotic, Department of Medical Sciences, Ministry of Public Health (Nonthaburi, Thailand). ICP multielement standard solution XIII was obtained from Agilent Technologies (Santa Clara, CA, USA). The suppliers for tryptone soya agar (TSA) and tryptic soy broth (TSB) were HiMedia Laboratories Private Limited (Maharashtra, India) and Becton, Dickinson and Company (Sparks, MD, USA), respectively. Folin–Ciocalteu was bought from Loba Chemie (Mumbai, India).

### 2.2. Preparation of Extracts

Single-factor experiments with the UAE method were used to investigate the effects of extraction conditions on the rhein content in *S. alata* leaf extracts. Each extraction was performed using an accurate weight (2.5 g) of dried powdered *S. alata* leaves (particle size 323.83 ± 2.42 µm) using an ultrasonicator (Model 230D, Crest Ultrasonics Corp., Ewing Township, NJ, USA) at a frequency of 42–45 kHz (level 9). As a first factor, the effects of various extraction solvents were examined. The herbal powders were separately extracted for 20 min at 50 °C using 50 mL of distilled water, 50% (*v*/*v*) ethanol and 95% (*v*/*v*) ethanol. For the subsequent extraction, the solvent with the highest rhein content was chosen. The concentration of the extraction solvent was the second factor affecting the rhein content in the extract. In this experiment, herbal powders were individually extracted with 50 mL of ethanol at various concentrations (60%, 70%, 80%, 90% and 95% (*v*/*v*)) for 20 min at 50 °C, and then the best solvent concentration was selected. The third factor was explored using various extraction temperatures. The powders were independently extracted for 20 min with 50 mL of 95% (*v*/*v*) ethanol at various temperatures (30 °C, 40 °C, 50 °C, 60 °C and 70 °C). The fourth factor was evaluated by varying the extraction times. Each powder sample was combined with 50 mL of 95% (*v*/*v*) ethanol and sonicated for 5 min, 10 min, 15 min, 20 min and 25 min at 50 °C. Finally, different solvent-to-solid ratios (10:1, 20:1, 30:1, 40:1 and 50:1) were used in a 15 min extraction at 50 °C. Following filtration through Whatman filter paper No. 1, each extraction solution was evaporated using a rotary evaporator (R-100, Buchi, Japan) operating at a pressure of 35 mbar and a temperature of 40 °C and was then completely dried using a freeze dryer (Model 6112974, Labconco Corporation, Kansas City, MO, USA). Prior to further analysis, all dried extracts were stored at −20 °C and kept in the dark.

Industrial and eco-friendly preparations of plant extract frequently reduce the number of organic solvents employed and the extraction temperature in order to minimize degradation of the active compounds. In order to compare the yield, bioactive content and biological activity of different extract types with the rhein-rich (optimized) extract, a variety of extracts were prepared under the conditions listed in Table 1.

### 2.3. Optimization of Extraction Conditions

To optimize the extraction conditions, a BBD comprising 17 experiments was used in conjunction with Design-Expert 8.0.6 software (Stat-Ease, Inc., Minneapolis, MN, USA). Three independent variables were selected in this design based on the results of single-factor analysis: extraction temperature, extraction time and solvent-to-solid ratio and their levels, and the response variable was rhein content, as depicted in Table 2 and Table 3. The extraction process was optimized with the goal of obtaining the highest level of rhein in *S. alata* leaf extracts.

### 2.4. Method for Chlorophyll Removal from Extracts

Rhein-rich extract was dissolved in 95% *v*/*v* ethanol and diluted in the same solvent to a concentration of 5 mg/mL. The solution was then combined separately with vegetable oils (palm oil or coconut oil) at a volume ratio of 1:1 or bleaching agents (aluminum oxide or activated charcoal) at a concentration of 150 mg/mL for 5 min using a vortex mixer. Using a centrifuge, the mixture was separated into two phases via centrifugation at 4500 rpm for 10 min at room temperature (Universal 320R, Hettich, North Rhine-Westphalia, Germany). The oil phase or precipitate was discarded, and the ethanolic phase was recovered. With certain adjustments, the chlorophyll removal efficiency was measured in accordance with previous work [38]. A UV-vis spectrophotometer (U-2900 spectrophotometer, Hitachi, Tokyo, Japan) was used to measure the absorbance values of the extract solutions at 652 nm before and after treatment. Each assay was accomplished in triplicate. Equation (1) was used to calculate the effectiveness of chlorophyll removal, as follows:Chlorophyll removal efficiency (%) = [(AbsB − AbsA)/AbsB] × 100 (1)
where AbsB is the extract’s absorbance before chlorophyll removal and AbsA is the extract’s absorbance after chlorophyll removal.

The resulting extracts were further evaluated for their total phenolic contents and biological activities.

### 2.5. Chromatographic Conditions and Method Development

The Agilent 1100 HPLC–DAD (Agilent Technologies, CA, USA) was used for all analyses. All chromatographic conditions were carried out with isocratic elution utilizing a Luna Omega Polar C18 column (5 μm, 100 Å, 4.6 mm × 250 mm) (Phenomenex Inc., Torrance, CA, USA), a flow rate of 1 mL/min and a detection wavelength of 254 nm, with some changes as detailed in Table 4. By comparing the unknown substance’s retention time and UV spectrum to the standard, rhein was found.

### 2.6. System Suitability Testing

The system suitability test was conducted as part of a routine to ensure the reliability of the validation results and to detect any alterations in the chromatographic equipment. The daily test comprised five runs of the same rhein standard solution with a specific concentration. Peak areas, retention durations, resolutions, symmetry factors and theoretical plates were all monitored, and the criteria followed the United States Pharmacopeia 43-National Formulary regulation 38 (USP 43-NF 38) [39].

### 2.7. Validation of Method

The analytical method that was employed to quantify rhein in *S. alata* leaf extracts was validated for specificity, linearity and range, accuracy, precision, limit of detection (LOD) and limit of quantitation (LOQ) in accordance with the International Conference on Harmonization (ICH) Q2 (R1) guideline [40].

Runs in triplicate with extract solution, rhein standard solution and a mobile phase blank were performed to demonstrate the method’s specificity. A calibration curve with concentrations ranging from 10 µg/mL to 30 µg/mL was created to test the linearity. For each concentration level, the experiment was duplicated three times. The LOQ and LOD were calculated using a statistical procedure that was applied to linearity data. The linear regression deviance and line slope were used to obtain the LOQ and LOD.

The ICH guidelines [40] indicate that the concentration range used to validate a method must be between 80% and 120% of the test concentration. The repeatability was tested by injecting 9 samples at 15, 20 and 25 μg/mL rhein. The same analyst performed all analyses on the same instrument on the same day. The intermediate precision was determined by repeating analyses in triplicate with three concentrations of the working range from the rhein standard solution on three separate days. The relative standard deviation (RSD) and Horwitz ratio (HorRat value) were estimated according to an earlier report [41].

An analysis of a sample solution spiked with a standard solution of rhein at three concentration levels of 75%, 100% and 125% was used to determine the accuracy. For each recovery level, samples were produced in triplicate, and injections were conducted in triplicate. The percentage of recovery was computed using Equation (2).
Recovery (%) = (Recovered concentration/Injected concentration) × 100(2)
where recovered concentration equals the concentration of the spiked sample minus the concentration of the non-spiked sample, and injected concentration represents the concentration of the standard rhein added.

Variations in the parameters of the oven temperature of the column with a fluctuation of ±2 °C and changes in the flow rate of the mobile phase with a variation of ±0.1 mL were assessed using robustness. In the same study, the quantification of rhein in samples was evaluated using standard rhein to assess parameter robustness.

To investigate chemical stability during storage, the rhein standard solution and extract sample solution were tested at room temperature for 0 and 36 h. The rhein’s chromatographic profile and peak area analysis were both verified.

### 2.8. Quantitative Determination of Rhein Contents

In this study, the rhein contents in *S. alata* leaf extracts were determined quantitatively using the established HPLC–DAD method. Individually, all extracts were dissolved and diluted to a concentration of 2 mg/mL with 95% *v*/*v* ethanol. All solutions were filtered with a 0.45 μm nylon syringe filter before analysis. Every experiment was carried out three times.

### 2.9. FTIR

Using a Fourier-transform infrared (FTIR) spectrophotometer (Nicolet Magna 4700, Thermo Fisher Scientific, Santa Clara, CA, USA), the functional groups of phytochemical components were studied. The dried extract sample (roughly 10 mg) was combined with potassium bromide (100 mg) and compacted using a hydraulic press to form a pellet of the sample. Each pellet was scanned in the 4000–400 cm^−1^ mid-infrared range.

### 2.10. Total Phenolic Contents

Total phenolic content was assessed according our previous study [21] with minor adjustments. An amount of 20 μL of the 50 μg/mL sample solution was mixed with 980 μL of Folin–Ciocalteu reagent and was let to stand for 5 min. After that, 1 mL of sodium carbonate solution (10% *w*/*v*) was added to the mixture. The reaction mixture was incubated at an ambient temperature for 60 min in the dark, and the absorbance was taken at 765 nm against a reagent blank using a Multimode microplate reader (Perkin-Elmer, Pontyclun, UK). Total phenolic content was evaluated using a gallic acid standard curve with concentrations ranging from 1.5 to 25 μg/mL and expressed as mg GAE/g extract (milligrams of gallic acid equivalents per gram of extract). Every analysis was carried out three times.

### 2.11. Determination of Minimal Inhibitory Concentrations (MIC) and Minimal Bactericidal Concentrations (MBC)

The Clinical Laboratory Standards Institute (CLSI)-recommended techniques for determining the MIC and MBC were used [42]. The MIC and MBC of the rhein-rich extract were determined using a two-fold standard broth dilution technique with an inoculum of approximately 10^7^–10^8^ CFU/mL against *Staphylococcus aureus* ATCC 6538P, *Escherichia coli* DMST 4212 and *Pseudomonas aeruginosa* ATCC 9027. A 1 mL stock solution of 20 mg/mL extract was combined and diluted two-fold with the test organism in TSB. Inoculated TSB (growth control), non-inoculated TSB (control) and a mixture of non-inoculated TSB and extract (negative control) were also tested. The extract’s MIC value was established as the lowest concentration that totally stopped bacterial growth after 24 h at 37 °C. To measure MBC, a quantity of liquid medium (15 μL) from each tube that showed no growth was taken and cultured for 24 h at 37 °C. After sub-culturing onto TSA plates, MBC was defined as the lowest concentration that exhibited no apparent bacterial growth. Cultures for control, positive and negative conditions were also created. Each assay was repeated at least three times.

### 2.12. Antioxidant Activities

The antioxidant activities of extract samples were evaluated using a modified version of the 2,2-diphenyl-1-picrylhydrazyl (DPPH) radical scavenging method [21,43]. With 95% *v*/*v* ethanol, DPPH was dissolved and diluted to a final concentration of 0.5 mM. The sample solution was diluted enough to provide a concentration of 0.15 to 2.5 mg/mL. In a 96-well microplate, 50 μL of the sample solution was combined with 150 μL of DPPH solution. The reaction mixture was kept at room temperature for 30 min in the dark before being measured at 515 nm using a Multimode microplate reader (Perkin-Elmer, Pontyclun, United Kingdom). All the measurements were taken in triplicate. The SC_50_ value for DPPH radical scavenging activity was estimated in terms of mean ± SD. Ascorbic acid, with concentrations from 2 to 30 μg/mL, was used as a positive control.

The ferric antioxidant power (FRAP) assay was based on a Fe^3+^-TPTZ complex being reduced to a Fe^2+^-TPTZ blue complex [21]. To make the freshly prepared FRAP reagent, 250 mL of 300 nM acetate buffer with a pH of 3.6, 25 mL of 10 nM TPTZ solution in 40 nM hydrochloric acid and 25 mL of 20 nM ferric chloride solution were combined. An amount of 20 μL of the sample solution was blended with 980 μL of FRAP solution in a 24-well microplate. The reaction mixture was held at 37 °C in the dark for 30 min before being read with a Multimode microplate reader at 593 nm (Perkin-Elmer, Pontyclun, UK). The ascorbic acid standard concentration ranged from 1 g/mL to 20 g/mL, and the sample concentration was 100 g/mL. All tests were carried out in triplicate. The antioxidant activity was measured in terms of FRAP values (milligrams of ascorbic acid equivalents per gram of extract).

### 2.13. Anti-Inflammatory Activities

The anti-inflammatory properties were tested using a modified version of the protein anti-denaturation assay described by Rick-Leonid et al. [44]. A 2 mL volume of fresh chicken egg albumin was combined with 28 mL of phosphate-buffered saline (PBS, pH 6.4), and then a 2 mL aliquot was mixed with 1.5 mL of the sample solution in a concentration range of 0.5–4.0 mg/mL. A negative control had a similar volume of PBS. After being incubated at 37 °C for 15 min, the mixtures were then heated at 70 °C for 5 min. After cooling, a U-2990 UV-vis spectrophotometer (Hitachi, Japan) was used to measure the absorbances at 660 nm. As a positive control, diclofenac diethylamine was used at concentrations of 0.25–2.5 μg/mL. The dose–response curve was used to determine the sample solution concentrations for 50% inhibition (IC_50_). Each sample was tested in triplicate.

### 2.14. Analysis of Heavy Metal Content

For quality control purposes, the heavy metal contents of the extracts were analyzed. As in our previous report [43], the concentrations of toxic elements such as arsenic (As), cadmium (Cd), lead (Pb) and mercury (Hg) in the optimized extract were determined via an Inductively Coupled Plasma Mass Spectrometer (ICP-MS) using microwave digestion. Approximately 1 g of the extract was digested in triplicate with 7 mL of a 65% *v*/*v* nitric acid solution using a microwave digester (Model ETHOS ONE, Milestone Corporation, Sorisole, Italy). Before being analyzed with an ICP-MS (Model 7500ce, Agilent Technologies, Santa Clara, CA, USA), the digestion solution was diluted to 25.0 mL with ultrapure water produced by the UltraPure Water Machine (TKA Wasseraufbereitungssysteme GmbH, Niederelbert, Germany). External calibration with five different concentrations of ICP multielement standard solution XIII in 5% *v*/*v* nitric acid solution was used to generate the calibration curve.

### 2.15. Microbial Limit Test

To ensure safety from pathogenic microorganism contamination, a microbial limit test was conducted. The total aerobic microbial count (TAMC), total yeast and mold count (TYMC), bile-tolerant Gram-negative bacteria, *Salmonella* spp., *E. coli* and *S. aureus* in the optimized extract were determined via a microbial enumeration test in accordance with USP 43-NF 38 [45]. For the determination of total aerobic microbial count using the plate method, the diluted extract solution (1 mL) was inoculated in 20 mL of Tryptic Soy Agar (TSA) and Sabouraud Dextrose Agar (SDA) and was then incubated at 32.5 ± 0.5 °C for 5 days and 25 ± 0.5 °C for 7 days, respectively. Within 1 h after preparing the appropriate dilution for inoculation, the sample was added to the medium. After incubation, the colonies on the plates were measured as colony forming units (CFU) per gram of extract. For the cultivation of other microbes, plates were incubated at 32.5 ± 0.5 °C for 24–48 h. The non-growing samples were plated onto selective agars and incubated at 32.5 ± 0.5 °C for 24–72 h to examine growth.

### 2.16. Statistical Analysis

Three replications of the data were used to calculate the mean and standard deviation (SD). Software called Statistics 9.0 was used to conduct the analysis. To assess whether there might be a difference between the means at *p* < 0.05, a one-way analysis of variance (ANOVA) and the least significant difference test were applied. By using Microsoft Excel 2017’s linear regression analysis, 50% inhibition was calculated.

## 3. Results and Discussions

### 3.1. The Development and Validation of the HPLC–DAD Method

#### 3.1.1. Screening of Optimized Conditions

The mobile phase composition, column temperature, and flow rate of the HPLC were all optimized (Table 4). In a reasonable amount of time, rhein (a bioactive marker) was separated using an isocratic elution. A 254 nm wavelength was chosen because UV light is absorbed by the aromatic chromophore of rhein (Figure 1). Better resolution, precision, accuracy and sensitivity in the separation of rhein were requirements for an effective chromatographic method. As shown in Table 4, several binary eluents were tested for the mobile phase composition using various ratios of organic solvents (such as methanol and acetonitrile) and weak acid aqueous solutions (such as acetic acid and phosphoric acid). Figure 2A–C show the HPLC chromatograms of *S. alata* leaf extract under various conditions of analysis. An improved ratio of 65:35 acetonitrile: 0.1% *v*/*v* aqueous phosphoric acid (condition C) showed better peak resolution and an elevated purity factor after several tests (Table 4). Peak broadening or tailing issues were also avoided by the mobile phase that was optimized.

The subsequent step involved validating a reverse-phase Luna Omega Polar C18 column (5 μm, 100 Å, 4.6 mm × 250 mm) with a flow rate of 1 mL/min and a detection wavelength of 254 nm using 65:35 acetonitrile: 0.1% *v*/*v* aqueous phosphoric acid as the mobile phase (condition C). The retention time of the rhein peak in this condition was 6.2 min (Figure 2C). The chromatograms of *S. alata* leaf extract spiked with rhein and standard rhein, analyzed under condition C, are shown in Figure 2D and Figure 2E, respectively. In comparison with previously published methods for the determination of rhein [14,15], the results of the present study reveal a shorter run time and lower solvent consumption. Additionally, this condition had the highest purity factor of 996.654 (Table 4).

#### 3.1.2. Method Validation

System suitability

In order to check the symmetry factor, theoretical plates and RSD of the peak area response, a suitability test was conducted on the chromatograms obtained under optimum circumstances. The results of the system suitability testing revealed the symmetry factor (0.85), theoretical plate (22453-22787) and RSD of the peak area response (0.20%), all of which fell within the USP 43-NF 38 regulation’s acceptable ranges of symmetry factors (0.8–1.5), theoretical plates (>2000), and RSD of the peak area response (≤2%) [39].

Specificity

Good separation from other components in the extract with the same retention time demonstrated the system’s specificity to rhein (Figure 2C,D). The purity of the rhein peak was assessed in the chromatograms of the extract and the extract spiked with standard rhein by comparing the superimposable UV spectra at the peak’s start, apex and end. According to Table 4, the purity factor and resolution were between 996 and 999 and 7.34 and 7.87, respectively, which were within the acceptable range. There was no interference with rhein’s peak. This demonstrated the method’s specificity.

Linearity

According to the Association of Official Agricultural Chemists (AOAC) guidelines [46], the developed method was found to be linear over a concentration range of 10 µg/mL to 30 µg/mL, and the coefficient of determination (*r*^2^) of standard rhein was 0.9990 (Table 5) for all replicates (*n* = 3), confirming the method’s linearity (y = 22.4138x + 13.2070).

Accuracy

For accuracy, a standard rhein solution with known concentrations (15, 20 and 25 μg/mL) was added to a pre-analyzed sample solution with a fixed concentration. According to Table 5, the developed method was found to be accurate because the high values of recovery (91.69–105.89%) fell within the acceptance criterion (80–110%) [46].

Precision

RSD values for repeatability (intra-day) and intermediate precision (inter-day) were 5.0% to 6.3% and 6.0% to 7.2%, respectively, and HorRat values were 0.8 to 0.9 and 0.6 to 0.7, respectively (Table 5). As per the acceptance criteria [46], the %RSD and HorRat values were ≤7.3% and 0.5–2.0, respectively, demonstrating that this method was found to be reasonably accurate.

LOD and LOQ

The signal-to-noise ratios used to estimate LOD and LOQ were 3.3 σ/S and 10 σ/S, respectively, where σ is the standard deviation of the response and S is the slope obtained from the calibration curve. The LOD was 2.44 μg/mL, and the LOQ was 7.39 μg/mL (Table 5).

Robustness

The parameters of the column oven temperature and mobile phase flow rate were assessed as part of the robustness evaluation. For each condition considered, samples were prepared in triplicate, and analyses were carried out in duplicate. In the assessment of the influence of the temperature variations in the column oven on the peak area responses, it was determined that the evaluated conditions of ±2 °C had no effect on the rhein analysis, indicating that the temperature in the robustness had no effect on the analysis. When the mobile phase flow rate was evaluated for its impact on the analytical outcomes, it was discovered that the evaluated parameter had no effect under the conditions of ±0.1 mL. The RSD values of rhein concentrations were less than 1.5% and 2.0%, respectively, for the robustness of column oven temperature and flow rate. The ANOVA with 95% confidence intervals yielded a *p*-value greater than 0.05, indicating that there was no significant difference in the data set, confirming that the method was robust under the conditions analyzed. Additionally, the robustness of the method was evaluated based on the resolution, symmetry factor and theoretical plate in relation to the rhein peak of the chromatographic separation. The resolution (6.70–7.00), symmetry factor (0.84–9.10) and theoretical plates (8965–9062) all met the robustness assessment’s acceptance criteria of resolution (>1.5), symmetry factor (0.8–1.5) and theoretical plates (>2000). The outcomes demonstrated the method’s robustness, with no alteration in rhein’s separation efficiency from the other peaks.

Chemical stability

At times 0 and 36 h, the *S. alata* extract samples were examined. The rhein concentration’s RSD values were 2.3 and 2.4 for 0 and 36 h, respectively. The stability data demonstrated that sample stability was maintained at room temperature for up to 36 h and that the variation obtained was acceptable.

These method validation outcomes showed that the developed method could be used effectively for the analysis of rhein in the samples of *S. alata* extract, which was in accordance with USP 43-NF 38 [39] and AOAC guidelines [46]. Furthermore, the developed and validated method had several advantages over the previous report [35], including reduced analysis time and organic solvent volume consumption.

### 3.2. Single-Factor Analysis

The independent variables and experimental range of the extraction conditions used in this study were meticulously chosen based on our previous research and other studies on the extraction of anthraquinones. In our previous study, 95% *v*/*v* ethanol extract contained more rhein than 50% *v*/*v* ethanol extract and aqueous extract [5]. Consequently, ethanol was chosen as the solvent. The ethanol concentration was set between 60% *v*/*v* and 95% *v*/*v*. According to Zhao et al. [47], the highest extraction yield of anthraquinones could be achieved at an extraction temperature of 67 °C for 33 min. The extraction yields decreased gradually as extraction temperatures surpassed 67 °C. The extraction times exceeding 33 min had no discernible effect on the extraction yield. Increasing extraction time might hasten the chemical breakdown of anthraquinones during the extraction process, resulting in a lower extraction yield. They proposed that an extraction temperature of 67 °C and an extraction time of 33 min are optimal conditions for obtaining the highest anthraquinone content using the UAE method. In our study, however, rhein content decreased after 25 min of extraction. Consequently, we selected an experimental temperature and time range of 30 °C to 70 °C and 5 min to 25 min, respectively. Wu et al. [48] reported that the extraction yields of anthraquinones significantly increased as the solvent-to-solid ratio increased from 20:1 mL/g to 26:1 mL/g and that the extraction temperature increased from 70 °C to 80 °C, especially at high extraction temperatures and a middle solvent-to-solid ratio. Hence, the solvent-to-solid ratio was between 10:1 mL/g and 50:1 mL/g.

In this study, preliminary trials were conducted to ascertain the independent key factors affecting the rhein content of *S. alata* leaf extracts and their respective levels. The type of extraction solvent, the solvent concentration, the extraction temperature, the extraction time and the solvent-to-solid ratio were all investigated as independent variables. The effects of different extraction solvents (distilled water, 50% *v*/*v* ethanol and 95% *v*/*v* ethanol) on the amount of rhein in the extracts are shown in Figure 3A, with 95% *v*/*v* ethanol being the best solvent. Figure 3B shows an increasing rhein content as the ethanol concentration increases, indicating that rhein is easily soluble in 95% *v*/*v* ethanol. These findings corroborate previous research indicating that rhein is a partially lipophilic anthraquinone with increased solubility in organic solvents [1]. As a result, 95% *v*/*v* ethanol was chosen as the extraction solvent. Figure 3C illustrates the effect of temperature on the rhein content, with the temperature increasing from 30 °C to 50 °C and then gradually decreasing. Increased temperature could assist in the extraction of rhein from *S. alata* powders by lowering the viscosity of the solvent, increasing the coefficient of diffusion and increasing the solubility of the active constituent [49]. As a consequence, the extraction temperature was set to 50 °C. The effect of extraction time on the rhein content of extracts is depicted in Figure 3D. It was discovered that the rhein content increased relatively slowly as the extraction time increased from 5 min to 15 min and then gradually decreased. Thus, the extraction time was set to 15 min. As shown in Figure 3E, rhein content steadily increases as the solvent-to-solid ratio increases from 10:1 mL/g to 30:1 mL/g, and the results stay unchanged thereafter. Consequently, a solvent-to-solid ratio of 30:1 mL/g was chosen as the optimization target.

### 3.3. Optimization of the Extraction Parameters of Rhein Content

The extraction temperature (A), the extraction time (B) and the solvent-to-solid ratio (C) were preferred for the optimization of the extraction conditions using the BBD based on the results of the single-factor analysis. Table 3 illustrates the pattern of the BBD and response results. Equation (3) for the rhein content was constructed using multiple regression analysis, as follows:Y = 9.64 +0.31*A +0.38*B −0.69*C +0.72*A*B +0.46*A*C − 0.35*B*C − 0.55*B^2^ − 0.94*C^2^(3)

Table 6 illustrates the ANOVA for the quadratic response surface model. This demonstrates that the mathematical model for rhein content is statistically significant with a *p*-value < 0.01. The *r*^2^ of the model is 0.9575. Moreover, there is no significant lack of fit at a *p*-value > 0.05. This establishes that the quadratic regression model adequately fits the data. A value of 0.8078 for “Pred R-Squared” is highly related to a value of 0.9149 for “Adj R-Squared,” indicating that the model is suitable for prediction. Similarly, at a *p*-value of <0.05, the linear effects of extraction temperature, time and solvent-to-solid ratio are statistically significant. Both the linear and quadratic effects are statistically significant.

Three-dimensional response surface plots and contour plots (Figure 4) were generated using the developed quadratic polynomial equation to illustrate the interaction between three independent variables and their effect on the rhein content. Various shades of blue to red indicate the increasing content of rhein. The effects of extraction temperature and time on the rhein content are depicted in Figure 4A,B when the solvent-to-solid ratio is constant. The yield of rhein increased as both factors increased, whereas a shorter extraction time and lower temperature resulted in a negligible amount of rhein. The interaction of these two variables had a significant effect on the rhein extraction from *S. alata* leaves. This finding is consistent with previous research on the optimization of rhein using UAE extracted from *Cassia fistula* pod pulp [50].

As illustrated in Figure 4C,D, the highest content of rhein was obtained using a medium solvent-to-solid ratio, a higher temperature and a constant extraction time. This could be because rhein is more soluble at a higher temperature. A lower temperature and high solvent-to-solid ratio are responsible for the lower rhein content. In fact, the solvent-to-solid ratio has a beneficial effect on the extraction process; as the solvent-to-solid ratio becomes higher, more phytochemicals are extracted. This is the principle of mass transfer; the driving force during mass transfer is the concentration gradient between the solid and the liquid’s bulk, which is greater when a higher solvent-to-solid ratio is used [51]. Nonetheless, a previous study reported that increasing the temperature and decreasing the solvent-to-solid ratio resulted in increased phytochemical concentrations. This is self-evident, as when less solvent is used, the phytochemical becomes more concentrated [52].

Figure 4E,F illustrate the effect of the interaction between extraction time and solvent-to-solid ratio. The highest content of rhein was obtained with a longer duration and a mild to moderate solvent-to-solid ratio. Increased solvent-to-solid ratios may result in rapid mass transfer between the solvent and plant materials. However, the amount of rhein was decreased in this study due to the high solvent-to-solid ratio, which is consistent with previous studies [48,53].

### 3.4. Optimal Extraction Conditions and Verification

As shown in Table 7, the optimal conditions for extraction were determined to be 59.52 °C, 18.4 min and a solvent-to-solid ratio of 25.48:1 (mL/g). Under these conditions, the software predicts a response value of 10.44 mg/g extract. Three additional experiments under optimal conditions were conducted, yielding an average experimental value of 10.36 mg/g extract. The observed value was within the predicted value’s 95% confidence interval. As a result, the model for the optimal conditions generated by the software could be validated and concluded to have a high degree of reliability.

### 3.5. Antibacterial Capacity of Rhein-Rich (Optimized) Extract

*E. coli*, *S. aureus* and *P. aeruginosa* strains were used to evaluate the antibacterial effect of the rhein-rich (optimized) extract because they are prevalent in chronic wounds and can cause infections under certain conditions, especially in diabetic patients with inadequate blood glucose control. The presence of *S. aureus* or anaerobes is significantly more problematic [54]. To validate the precision and dependability of the experimental measurements, neomycin sulfate was used as a positive control. The rhein-rich extract demonstrated antibacterial activity against the three strains, whose respective MICs for *S. aureus*, *E. coli*, and *P. aeruginosa* were 0.625 mg/mL, 1.25 mg/mL and 1.25 mg/mL, as shown in Table 8.

According to research by Douhari J.H. and Okafor B [55], the MICs of the methanolic extract of *S. alata*’s roots and leaves against *S. aureus*, *E. coli* and *P. aeruginosa* were 12 mg/mL, 15 mg/mL and 20 mg/mL, respectively. Furthermore, Ehiowemwenguan et al. [56] found that the MICs of the methanolic extract of the leaves of *S. alata* against *S. aureus*, *E. coli* and *P. aeruginosa* were 6 mg/mL, 8 mg/mL and 10 mg/mL, respectively. Pham et al. demonstrated that bioassay-guided fractionation led to the identification of antimicrobial components from *S. alata* leaf extract. Antimicrobial rhein was abundant in extracts of *S. alata* leaves soluble in methanol and ethyl acetate [19]. According to our findings, the optimized extract possessed antimicrobial properties against Gram-positive (*S. aureus*) and Gram-negative (*E. coli* and *P. aeruginosa*) microorganisms. According to a previous study [57], optimized *S. alata* leaf extract could be a suitable alternative to antibiotics for treating chronic wounds. In this study, the rhein-rich extract had the lowest MIC against *S. aureus*. The next step, which entailed comparing and decolorizing the extracts, was to test for antibacterial activity against *S. aureus*.

### 3.6. Evaluation of S. alata Leaf Extracts

According to Table 9, the best extraction solvent was distilled water, which provided the highest yield (19.03 ± 0.15%) of all *S. alata* leaf extracts. For the polar phytochemical components, warm water worked well as an extraction solvent but not for rhein and phenolic compounds. Although the hydroalcoholic solvents (50% *v*/*v* ethanol and 95% *v*/*v* ethanol) did not increase yields, they did increase the rhein and phenolic contents of *S. alata* leaf extracts. This indicates that the decreased dielectric constant of the extraction solvents plays a significant role in UAE’s increased rhein and phenolic concentrations.

An increase in rhein concentration was always accompanied by an increase in total phenolic content (Table 9). The MIC and MBC are better, and the rhein and total phenolic contents are higher. The rhein-rich extract obtained from the UAE method under ideal extraction conditions had the highest concentrations of rhein (10.36 ± 0.27 mg/g extract) and total phenolic content (105.75 ± 0.97 mg GAE/g extract), as well as the most potent antibacterial activity against *S. aureus* (MIC of 0.625 mg/mL and MBC of 5.0 mg/mL). The results demonstrated that a decreasing dielectric constant with increasing temperature and solvent: solid ratio improved rhein content, total phenolic content and antibacterial activity (Table 1 and Table 9).

In comparison to an earlier work [58], where extract was obtained via maceration with 70% *v*/*v* aqueous methanol, our findings optimized the solvent type (95% *v*/*v* aqueous ethanol) and extraction method (UAE) employed for the creation of *S. alata* leaf extracts. Ethanol was a better solvent than methanol because it was less poisonous and could produce high rhein and phenolic extracts with antibacterial properties. As previously reported [59,60], the antibacterial activity of *S. alata* leaf extracts might well be due to the presence of rhein and phenolic compounds, consistent with our findings.

### 3.7. Evaluation of De-Chlorophyll Extracts

The rhein-rich (optimized) extract was superior to the other extracts because it contained the highest levels of rhein and phenolics and had the strongest antibacterial activity (Table 9). Despite this, the rhein-rich extract retained its dark brown hue. In order to decolorize it, bleaching agents or liquid–liquid extraction with vegetable oils were utilized. Table 10 displays the decolorizing agents and their chlorophyll removal efficiency. Activated charcoal had the best chlorophyll removal efficiency (92.87 ± 0.20%), followed by aluminum oxide (57.41 ± 0.09%), coconut oil (25.43 ± 0.08%) and palm oil (17.33 ± 0.05%). De-chlorophyll extracts were used in additional research to determine FTIR spectra, total phenolic contents and biological activities.

As illustrated in Figure 5, the FTIR spectra of the rhein-rich extract and its de-chlorophyll extracts with palm oil, coconut oil, aluminum oxide and activated charcoal displayed broad bands of hydroxyl groups (at 3100–3600 cm^−1^), medium peaks of C-H stretching (at 2900–2950 cm^−1^), pairs of C=C-C stretching bands in the aromatic rings (at 1617–1620 cm^−1^ and 1452–1455 cm^−1^), strong peaks of conjugated carbonyl groups (at 1695–1700 cm^−1^) and medium peaks of C-O groups (at 1068–1074 cm^−1^). The presence of these particular peaks/bands in the spectra was consistent with the structure of rhein (Figure 1) and previous findings [60], indicating that rhein was the principal component of these extracts.

As the DPPH SC_50_ value became lower and the FRAP value became higher, the antioxidant activity of the sample became greater [61]. As the IC_50_ became lower, the anti-inflammatory activity in albumin denaturation became greater [62]. Although aluminum oxide and activated charcoal were able to effectively reduce the chlorophyll concentration in rhein-rich extracts with yellow solutions (Table 10 and Figure 6), rhein content, total phenolic content, antioxidant activity, anti-inflammatory activity and antibacterial activity against *S. aureus* were all decreased (Table 11). Conversely, the decolorization step with vegetable oils (palm oil or coconut oil) increased the total phenolic content and antioxidant activity (Table 11) and provided an acceptable physical appearance (a lighter brown solution) (Figure 6).

As shown in Table 11, the rhein-rich extract decolorized with coconut oil had the highest total phenolic content and the strongest antioxidant activity (the lowest SC_50_ and the highest FRAP value), whereas the rhein-rich extract decolorized with activated charcoal had the strongest anti-inflammatory activity (the lowest IC_50_). The increase in phenolic content and antioxidant activity of de-chlorophyll extracts might be attributable to the solubilization of phenolic compounds from vegetable oils in an organic solvent (95% *v*/*v* aqueous ethanol) during liquid–liquid extraction. According to a previous report [63], coconut oil contains fewer total phenolics than palm oil. Phenolics play an antioxidant role via chain reactions of free radicals, which has a substantial effect on the antioxidant activity of oils [64]. However, de-chlorophyll extract with coconut oil had a significantly higher total phenolic content and antioxidant activity than the extract with palm oil. This might be attributable to the high solubility of phenolics from coconut oil and the presence of phenolics with high antioxidant activity, which are predominant in coconut oil [65].

Although previous studies [1,2] have illustrated that rhein possessed antioxidant and anti-inflammatory properties, this study found that the rhein content in de-chlorophyll extracts correlated directly with their anti-inflammatory potency (Table 11). In addition, the removal of chlorophyll from rhein-rich extracts with various decolorizing agents had varying effects on rhein content reduction. Results show that chlorophyll removal by bleaching agents (aluminum oxide or activated charcoal) had a smaller effect on rhein content than liquid–liquid extraction with vegetable oil (palm oil or coconut oil). However, de-chlorophyll extracts treated with vegetable oils had a greater total phenolic content and stronger antioxidant activity. Consequently, it was possible that the content of phenolic compounds in the extract had a big effect on its antioxidant activity, and the content of rhein in the extract had a big effect on its anti-inflammatory activity. The findings of this study are consistent with those of previous studies [66,67].

Due to the highest concentration of rhein and the strongest antibacterial activity (the lowest MIC value) against *S. aureus* (Table 7 and Table 8) of the rhein-rich (optimized) extract, rhein was selected as a bioactive marker compound for the investigation of the antibacterial activities of de-chlorophyll extracts (Table 11). The MIC values of rhein-rich and de-chlorophyll extracts against *S. aureus* varied between 0.625 mg/mL and 2.5 mg/mL. Analysis of the antibacterial activity of *S. alata* extracts revealed that the rhein-rich extract exhibited excellent antibacterial activity against *S. aureus* and that the antibacterial activity (MIC) of *S. alata* extracts was positively correlated with the rhein content, indicating that rhein was a significant antibacterial component in *S. alata* extract against the growth of *S. aureus*. According to a previous study [68], the antibacterial activity of rhein, an anthraquinone derivative, might be related to the hydroxyanthraquinone nucleus consisting of two ketone groups at positions 9 and 10 and two hydroxyl groups at positions 1 and 8. In addition, rhein has a substituted polar carboxylic group at position 3 (Figure 1), which can enhance its antibacterial activity. This study demonstrated that the rhein content of *S. alata* extracts had a beneficial effect on antibacterial activity against *S. aureus*.

The presence of an ionizable carboxylic group on the rhein molecule (Figure 1) favored interaction with the aluminum oxide adsorbent, according to previous research [69]. Activated charcoal is graphite with deformations. The chemical structure of activated charcoal resembles graphite. The layers of graphite crystal are held together by van der Waals forces [70]. Carbon–carbon bonds hold layers together. The interaction between rhein and activated charcoal can be interpreted as the intercalation of rhein into the interlayer spaces of sorbents. The fact that rhein has a planar geometry of an anthracene ring (a tricyclic aromatic ring) (Figure 1) is highly favorable for its effective intercalation into activated charcoal [69,70]. Due to the weak interactions between the rhein molecule and adsorbents, the rhein content in de-chlorophyll extracts treated with aluminum oxide or activated charcoal was marginally reduced (Table 11).

Based on the results of the experiment, it was determined that activated charcoal was the most effective for removing chlorophyll (chlorophyll removal efficacy = 92.87 ± 0.20%) (Table 10), resulting in the lightest yellow color (Figure 6). Additionally, the de-chlorophyll extract treated with activated charcoal retained the maximum content of rhein (8.52 ± 0.08 mg/g extract) and exhibited the strongest antibacterial activity (the lowest MIC = 1.25 mg/mL) and anti-inflammatory activity (the lowest IC_50_ = 1.94 ± 0.04 mg/mL) (Table 11).

### 3.8. Heavy Metal Contents in the Rhein-Rich (Optimized) Extract

Pb and Cd have adverse bimolecular functional effects, primarily neurodegeneration, whereas minimal levels of As and Hg affect the pulmonary, neurological, renal and respiratory systems [71]. Thus, it is essential to investigate heavy metal risk in herbal extracts. To ensure its safety, the levels of As, Cd, Pb and Hg in the rhein-rich extract were determined using ICP-MS. The ICP-MS calibration curves exhibited excellent linearity, with *r^2^* greater than 0.9995. As shown in Table 12, the concentrations of these metals in the extract were well below the contamination limits established by the Association of Southeast Asian Nations (ASEAN) [72]. This study demonstrated that rhein-rich extract poses no threat to human health.

### 3.9. Microbial Contamination in Rhein-Rich (Optimized) Extract

According to the ASEAN guidelines on the limits of contaminants [72], the acceptable criteria for microbiological quality based on TAMC, TYMC, bile-tolerant Gram-negative bacteria and specified microorganisms (including *Salmonella* spp., *E. coli* and *S. aureus*) must not exceed 2 × 10^4^ CFU/g, 2 × 10^2^ CFU/g, 10^2^ CFU/g and the absence of specified bacteria in the given sample weight, respectively. The present study’s results (Table 13) demonstrate that the rhein-rich extract and that stored at 4 °C for 6 months met the criteria set. Consequently, the microbiological quality of the rhein-rich extract was of exceptional quality.

## 4. Conclusions

This study validated the rhein assay method established in accordance with the ICH Q2 (R1) guideline. The analysis was performed using a developed HPLC–DAD method with a Luna Omega Polar C18 column (5 μm, 100 Å, 4.6 mm × 250 mm). The mobile phase was acetonitrile and water containing 0.1% *v*/*v* of phosphoric acid (65:35) at a flow rate of 1 mL/min, and the detector was set at 254 nm. Rhein was separated with a good resolution value of 7.34 to 7.87, a high purity factor of 996 to 999 and a relatively short retention time of 6.2 min. The determination of rhein in *S. alata* leaf extracts was successful using this validated method. Likewise, it reduced the amount of time and solvent required for analysis. This is the first publication describing a fully validated HPLC–DAD method for rhein, used as a quality control marker in *S. alata* leaf extracts.

For preparing the rhein-rich extract of *S. alata* leaves using the UAE method, the ideal conditions were 95% *v*/*v* ethanol as the solvent, a solvent-to-solid ratio of 25: 1 (mL/g), a temperature of 60 °C and an extraction time of 18 min. Compared to other obtained extracts, the rhein-rich (optimized) extract had the highest content of rhein and phenolics, as well as the most potent antibacterial activity against *S. aureus*.

Nonetheless, the rhein-rich extract remained dark brown in color. In order to decolorize this extract, eco-friendly and efficient techniques such as bleaching agents (activated charcoal or aluminum oxide) and liquid–liquid extraction with vegetable oils (palm oil or coconut oil) were employed. Each de-chlorophyll extract appeared lighter in color. FTIR spectroscopy revealed that all de-chlorophyll extracts still contained rhein. The results demonstrate that chlorophyll removal from the rhein-rich extract via vegetable oil extraction produced de-chlorophyll extracts with a higher phenolic content and enhanced antioxidant activity. In addition, the removal of chlorophyll from the rhein-rich extract via adsorption with bleaching agents resulted in the production of de-chlorophyll extracts with high levels of rhein and good anti-inflammatory and antibacterial activities.

In addition to their antioxidant, anti-inflammatory and antibacterial properties, however, more research should be conducted on the bioactivities of de-chlorophyll extracts derived from rhein-rich extract. As a result of these findings, the preparation of de-chlorophyll rhein-rich *S. alata* extracts with antioxidant, anti-inflammatory and antibacterial activities, along with the validation of a rhein determination method, could be applied to the development of herbal cosmeceuticals and pharmaceuticals containing *S. alata* leaf extract.

## Figures and Tables

**Figure 1 antibiotics-12-00181-f001:**
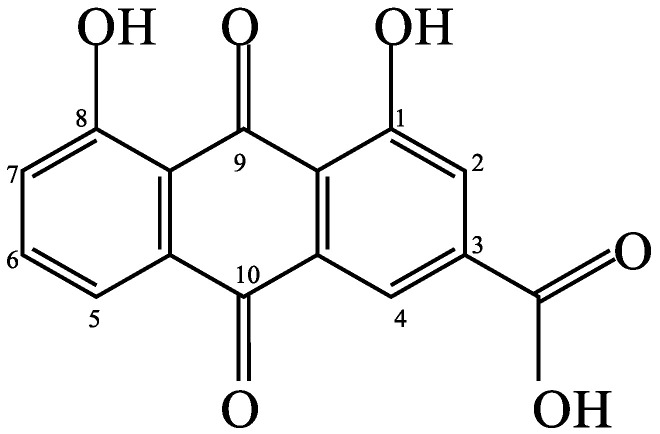
Structure of rhein.

**Figure 2 antibiotics-12-00181-f002:**
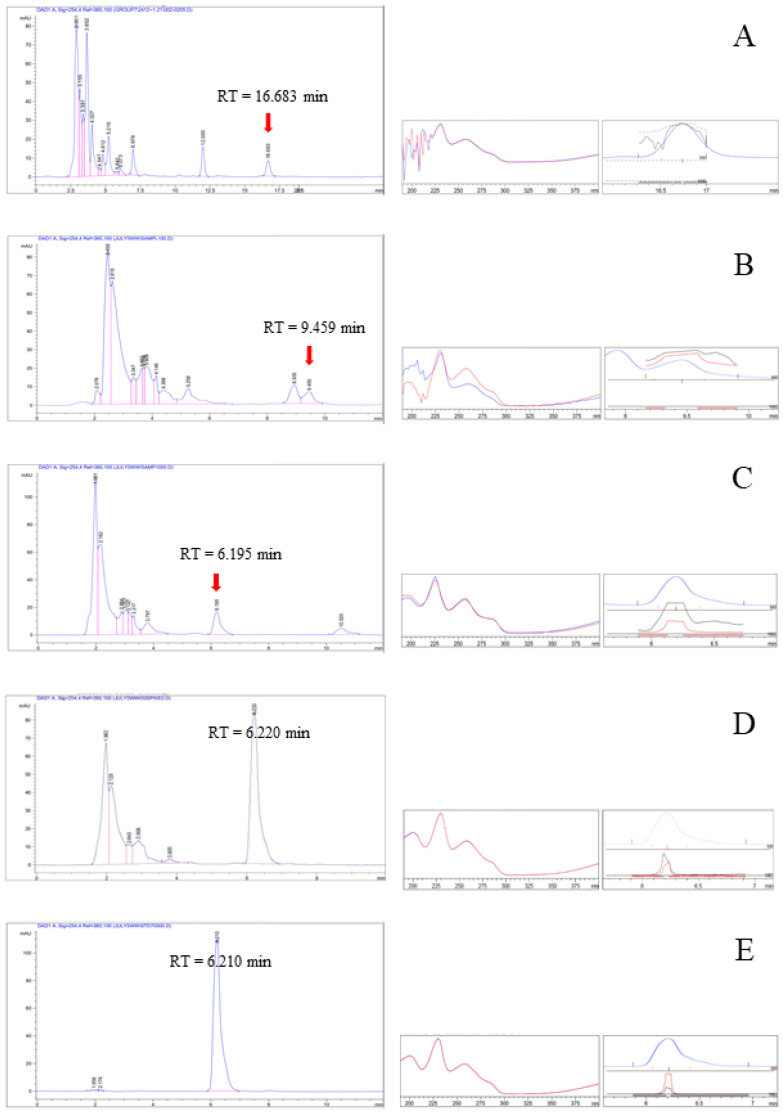
HPLC chromatograms of *S. alata* leaf extract analyzed under condition A (**A**), condition B (**B**) and condition C (**C**) and with extract spiked with rhein analyzed under condition C (**D**) and standard rhein analyzed under condition C (**E**) with a 254 nm detection wavelength.

**Figure 3 antibiotics-12-00181-f003:**
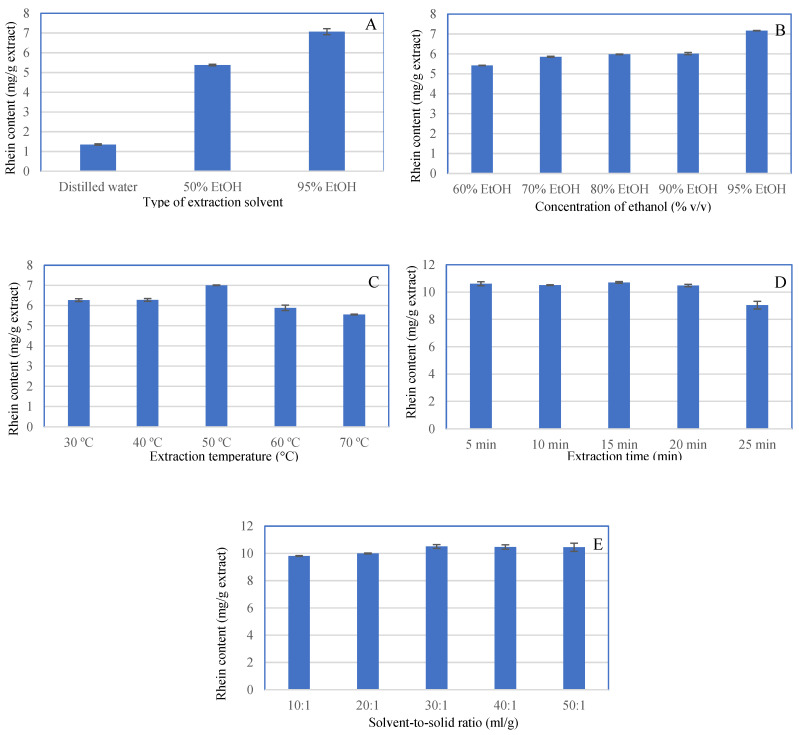
Effects of solvent type (**A**), solvent concentration (**B**), extraction temperature (**C**), extraction time (**D**) and solvent-to-solid ratio (**E**) on the rhein content of the extracts.

**Figure 4 antibiotics-12-00181-f004:**
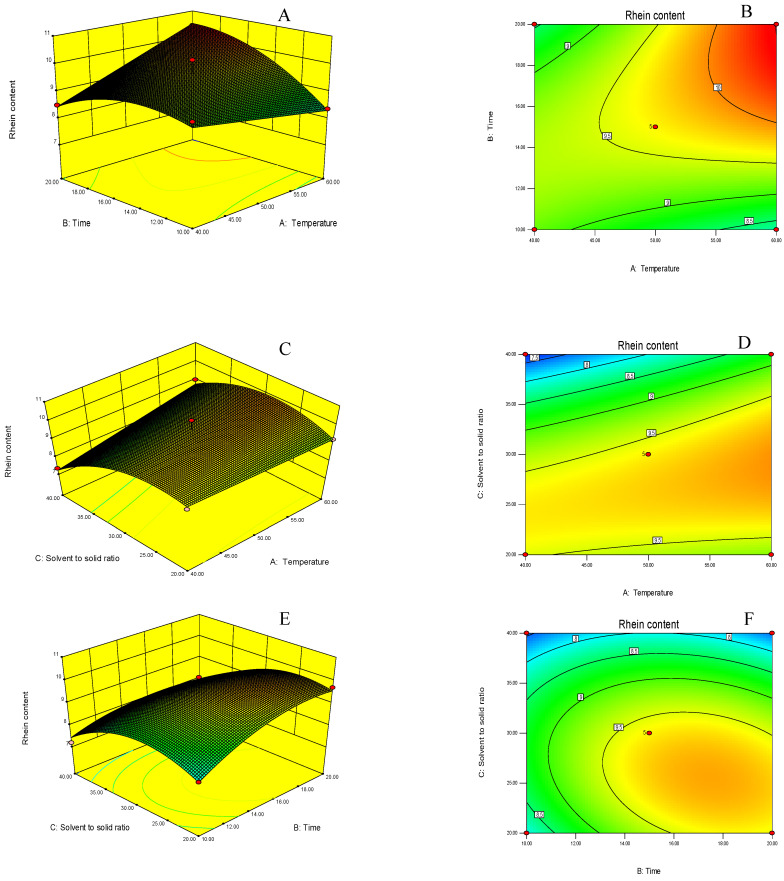
Three-dimensional response surface and contour plots: temperatures and extraction times (**A**,**B**), temperatures and solvent-to-solid ratios (**C**,**D**) and extraction times and solvent-to-solid ratios (**E**,**F**) on rhein content.

**Figure 5 antibiotics-12-00181-f005:**
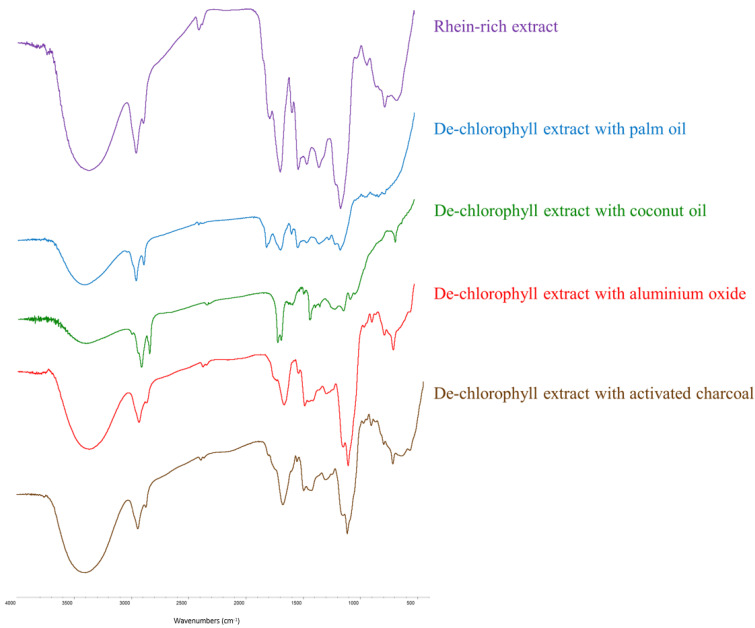
FTIR spectra of rhein-rich extract and its de-chlorophyll extracts with palm oil, coconut oil, aluminum oxide and activated charcoal.

**Figure 6 antibiotics-12-00181-f006:**
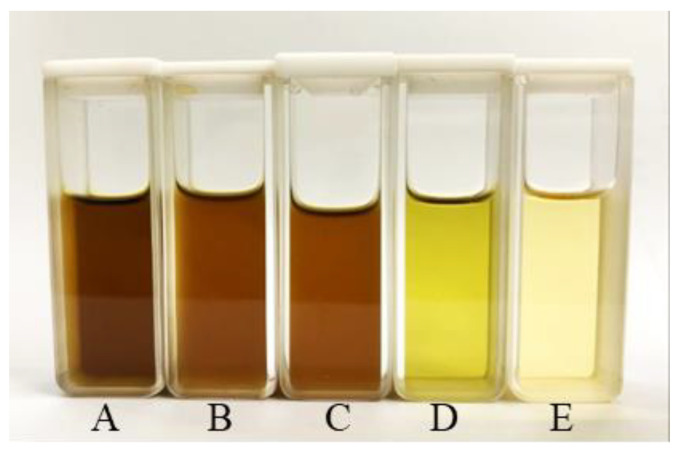
Solution of rhein-rich extract (**A**) from *S. alata* leaves with an intense brownish color decolorized using palm oil (**B**), coconut oil (**C**), aluminum oxide (**D**) and activated charcoal (**E**) as the decolorizing agents.

**Table 1 antibiotics-12-00181-t001:** Names of extract samples indicating conditions of extraction.

Name of Extract	Solvent	Solvent:Solid Ratio (mL/g)	Temperature (°C)	Time (min)
Aqueous	Aqueous extract	20:1	50	20
50% EtOH	50% *v*/*v* Ethanol	20:1	50	20
95% EtOH	95% *v*/*v* Ethanol	20:1	50	20
Rhein-rich (Optimized)	95% *v*/*v* Ethanol	25:1	60	18

**Table 2 antibiotics-12-00181-t002:** Variables and factor levels for optimization of extraction.

Independent Variables	Symbol	Levels		
		−1	0	1
Temperature (°C)	A	40	50	60
Time (min)	B	10	15	20
Solvent-to-solid ratio (mL/g)	C	20	30	40

**Table 3 antibiotics-12-00181-t003:** Coded (actual) levels of the operational parameters and observed values of BBD.

No.	Temperature (°C) (A)	Extraction Time (min) (B)	Solvent-to-Solid Ratio (mL/g) (C)	Rhein Content (mg/g Extract)
1	0 (50)	0 (15)	0 (30:1)	9.49
2	0 (50)	−1 (10)	−1 (20:1)	8.12
3	+1 (60)	−1 (10)	0 (30:1)	8.36
4	0 (50)	0 (15)	0 (30:1)	9.45
5	+1 (60)	+1 (20)	0 (30:1)	10.42
6	−1 (40)	+1 (20)	0 (30:1)	8.50
7	−1 (40)	−1 (10)	0 (30:1)	9.31
8	−1 (40)	0 (15)	+1 (40:1)	7.35
9	+1 (60)	0 (15)	+1 (40:1)	9.02
10	0 (50)	+1 (20)	+1 (40:1)	7.39
11	0 (50)	−1 (10)	+1 (40:1)	7.19
12	+1 (60)	0 (15)	−1 (20:1)	9.24
13	0 (50)	0 (15)	0 (30:1)	10.15
14	−1 (40)	0 (15)	−1 (20:1)	9.39
15	0 (50)	0 (15)	0 (30:1)	9.35
16	0 (50)	+1 (20)	−1 (20:1)	9.71
17	0 (50)	0 (15)	0 (30:1)	9.58

**Table 4 antibiotics-12-00181-t004:** Analytical conditions of the HPLC system for determining rhein.

Parameters	Chromatographic Conditions		
	A	B	C
Column	Luna Omega Polar C18 column (5 μm, 100 Å, 4.6 mm × 250 mm) (Phenomenex Inc., Torrance, CA, USA)		
Column temperature	25 °C	30 °C	30 °C
Flow rate	1 mL/min	0.8 mL/min	1 mL/min
Injection volume	10 μL	10 μL	10 μL
Detection wavelength	254 nm	254 nm	254 nm
Mobile phase	Methanol: 2% *v*/*v* aqueous acetic acid (70:30)	Acetonitrile: 0.5% *v*/*v* aqueous acetic acid (60:40)	Acetonitrile: 0.1% *v*/*v* aqueous phosphoric acid (65:35)
Run time	20 min	15 min	15 min
Resolution	10.79	0.88	6.80
Symmetry factor	0.85	1.20	0.85
Theoretical plate	17,520	3315	8968
Retention time (min)	16.683	9.459	6.195
Purity factor	238.174	893.232	996.654

**Table 5 antibiotics-12-00181-t005:** Summary of analytical method validation.

Validation Parameters	Acceptance Criteria *	Results
Specificity	Resolution ≥ 2	Resolution = 7.34–7.87
	Purity factor ≥ 950	Purity factor = 996–999
Linearity and range	*r*^2^ ≥ 0.99	*r*^2^ = 0.9990
Accuracy	Recovery = 80–110%	Recovery = 91.69–105.89%
Repeatability (Intra-day precision)	RSD ≤ 7.3%	RSD = 5.0–6.3%
	HorRat value = 0.5–2.0	HorRat value = 0.8–0.9
Intermediate precision (Inter-day precision)	RSD ≤ 7.3%	RSD = 6.0–7.2%
	HorRat value = 0.5–2.0	HorRat value = 0.6–0.7
Robustness	Robust	Robust
Limit of detection	-	2.44 µg/mL
Limit of quantitation	-	7.39 µg/mL

* AOAC, 2016 [46].

**Table 6 antibiotics-12-00181-t006:** ANOVA of the quadratic model for rhein contents of *S. alata* leaf extracts.

Source	Sum of Squares	df	Mean Square	F Value	*p* Value	Remarks
Model	14.39	8	1.80	22.51	0.0001	**
A	0.77	1	0.77	9.68	0.0144	*
B	1.16	1	1.16	14.50	0.0052	**
C	3.78	1	3.78	47.36	0.0001	**
AB	2.06	1	2.06	25.83	0.0010	**
AC	0.83	1	0.83	10.39	0.0122	*
BC	0.49	1	0.49	6.08	0.0389	*
B^2^	1.27	1	1.27	15.84	0.0041	**
C^2^	3.77	1	3.77	47.16	0.0001	**
Residual	0.64	8	0.080			
Lack of fit	0.24	4	0.060	0.60	0.6854	Not significant

* 0.01 < *p* < 0.05, ** *p* < 0.01.

**Table 7 antibiotics-12-00181-t007:** Predicted and actual values for the verification of optimal extraction conditions.

	Temperature (°C)	Time (min)	Solvent-to-Solid Ratio (mL/g)	Rhein Content (mg/g Extract)
Optimal condition	59.52	18.4	25.48	10.44 (Predicted value)
Experimental condition	60	18	25	10.36 ± 0.27 (Observed value)

**Table 8 antibiotics-12-00181-t008:** Minimum inhibitory concentration (MIC) values of rhein-rich extract.

Tested microorganisms	Optimized Extract		Neomycin Sulfate	
	MIC (mg/mL)	MBC (mg/mL)	MIC (mg/mL)	MBC (mg/mL)
*Staphylococcus aureus* ATCC 6538P	0.63	5.0	0.063	0.125
*Escherichia coli* DMST 4212	1.25	2.5	0.016	0.031
*Pseudomonas aeruginosa* ATCC 9027	1.25	5.0	0.031	0.250

**Table 9 antibiotics-12-00181-t009:** Yields, rhein contents, total phenolic contents, MIC and MBC against *S. aureus* in *S. alata* leaf extracts.

Name of extract	Yield (%)	Rhein Content (mg/g Extract)	Total Phenolic Content (mg GAE/g Extract)	MIC (mg/mL)	MBC (mg/mL)
Aqueous	19.03 ± 0.15 ^d^	2.03 ± 0.01 ^a^	32.55 ± 3.18 ^a^	5.0	40
50% EtOH	14.65 ± 0.23 ^c^	6.83 ± 0.10 ^b^	55.98 ± 2.03 ^b^	2.5	40
95% EtOH	8.45 ± 0.35 ^a^	7.81 ± 0.12 ^c^	98.75 ± 1.26 ^c^	1.25	10
Rhein-rich (Optimized)	11.35 ± 0.45 ^b^	10.36 ± 0.27 ^d^	105.75 ± 0.97 ^d^	0.63	5.0

Different letters in the same column represent statistically significant differences with a confidence interval of 95% (*p* < 0.05). MIC = minimum inhibitory concentration, MBC = minimum bactericidal concentration.

**Table 10 antibiotics-12-00181-t010:** Decolorizing agents and their chlorophyll removal efficacy.

Decolorizing Agents	Extract Solution/Oil Volume Ratio or Decolorizing Agent Concentration	Chlorophyll Removal Efficacy (%)
Palm oil	1:1 (*v*/*v*)	17.33 ± 0.05 ^a^
Coconut oil	1:1 (*v*/*v*)	25.43 ± 0.08 ^b^
Aluminum oxide	150 mg/mL	57.41 ± 0.09 ^c^
Activated charcoal	150 mg/mL	92.87 ± 0.20 ^d^

Mean ± SD, *n* = 3. Significant differences exist between values in the column with different super-script characters (*p* < 0.05).

**Table 11 antibiotics-12-00181-t011:** Rhein content, total phenolic content, antioxidant activity, anti-inflammatory activity and antibacterial activity against *S. aureus* of rhein-rich extract and its de-chlorophyll extracts.

Samples	Rhein Content (mg/g Extract)	Total Phenolic Content (mg GAE/g Extract)	Scavenging Activity against DPPH, SC_50_	FRAP Value (mg AAE/g Extract)	Albumin Denaturation Activity, IC_50_ (mg/mL)	MIC (mg/mL)
Rhein-rich extract	10.36 ± 0.27 ^e^	105.75 ± 0.97 ^c^	1.25 ± 0.12 mg/mL ^c,d^	112.35 ± 1.87 ^c^	1.81 ± 0.03 ^b^	0.625
De-chlorophyll extract with various decolorizing agents						
Palm oil	1.50 ± 0.10 ^b^	107.30 ± 3.46 ^d^	1.17 ± 0.08 mg/mL ^c^	124.59 ± 1.96 ^d^	2.51 ± 0.04 ^d^	2.5
Coconut oil	1.42 ± 0.09 ^a^	120.43 ± 2.01 ^e^	1.01 ± 0.12 mg/mL ^b^	156.65 ± 3.57 ^e^	2.64 ± 0.03 ^e^	2.5
Aluminum oxide	7.98 ± 0.02 ^c^	27.26 ± 0.38 ^a^	1.43 ± 0.07 mg/mL ^d^	44.34 ± 1.58 ^b^	2.01 ± 0.05 ^c^	1.25
Activated charcoal	8.52 ± 0.08 ^d^	34.97 ± 5.91 ^b^	1.69 ± 0.10 mg/mL ^e^	38.29 ± 0.18 ^a^	1.94 ± 0.04 ^b, c^	1.25
Ascorbic acid	n/a	n/a	15.06 ± 0.37 µg/mL ^a^	n/a	n/a	n/a
Diclofenac diethylamine	n/a	n/a	n/a	n/a	1.04 ± 0.09 ^a^	n/a
Neomycin sulfate	n/a	n/a	n/a	n/a	n/a	0.0625

Values expressed as mean ± SD. Different letters in the same column represent statistically significant differences with a confidence interval of 95% (*p* < 0.05). n/a: not applicable.

**Table 12 antibiotics-12-00181-t012:** Heavy metal concentrations in rhein-rich extract.

Heavy metal	Limit of Detection * (mg/kg)	Permissible Limit (mg/kg)	Observed Values (mg/kg)
As	<0.041	5.0	ND
Cd	<0.021	0.3	ND
Pb	<0.020	10.0	ND
Hg	<0.054	0.5	ND

* As per limits mentioned in ASEAN guidelines. ND = Not detected; Less than the limit of detection.

**Table 13 antibiotics-12-00181-t013:** Microbiological quality of the rhein-rich extract and its 6-month storage at 4 °C.

Microbiological Quality	Acceptable Criteria *	Rhein-Rich Extract	6-Month Storage Extract
Total aerobic microbial count (TAMC)	NMT 2 × 10^4^ CFU/g	Absence in 1 g	10 CFU/g
Total yeast and mold count (TYMC)	NMT 2 × 10^2^ CFU/g	Absence in 1 g	Absence in 1 g
Bile-tolerant Gram-negative bacteria	NMT 10^2^ CFU/g	Absence in 1 g	Absence in 1 g
*Salmonella* spp.	Absence in 10 g	Absence in 10 g	Absence in 10 g
*Escherichia* coli	Absence in 1 g	Absence in 1 g	Absence in 1 g
*Staphylococcus aureus*	Absence in 1 g	Absence in 1 g	Absence in 1 g

* As per limits mentioned in ASEAN guidelines. NMT = Not more than.

## Data Availability

The data presented in this study are contained within the article.

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
