# Peer review of "Preparation of Bioactive De-Chlorophyll Rhein-Rich Senna alata Extract"

_antibiotics, 2023, doi:10.3390/antibiotics12010181_

Round 1

Reviewer 1 Report

I have carefully read the manuscript and find it very interesting and significant for the scientific community. The manuscript has been prepared very well in clear English with good discussion. However, some issues need to be addressed before further consideration for publication:

1. How did you choose the conditions for ultrasound-assisted extraction? Did you use available literature or conduct preliminary experiments? If you conduct preliminary experiments, results should be presented.

2. Please provide the chromatogram of the extract obtained under optimal conditions.

3. Why did you analyze only heavy metals? What about other elements which are significant for human health?

Reviewer 2 Report

Dear Authors

Overall, the publication is beautifully designed. However, some technical and minor corrections are required. Notes have been made in the pdf file.

Reviewer 3 Report

Taking into account that the wording "De-chlorophyll extracts" is not common in the specialized literature, I would suggest that this term be explained at the beginning of the paper.

Round 2

Reviewer 1 Report

I have carefully checked the revised manuscript and the authors' answers. The authors answered all issues successfully. Therefore, I suggest acceptance of the manuscript in its current form.